# A highly responsive pyruvate sensor reveals pathway-regulatory role of the mitochondrial pyruvate carrier MPC

Robinson Arce-Molina[1,2], Francisca Cortés-Molina[1], Pamela Y Sandoval[1], Alex Galaz[1], Karin Alegría[1], Stefanie Schirmeier[3], L Felipe Barros[1]*, Alejandro San Martín[1]*

[1]Centro de Estudios Científicos-CECs, Valdivia, Chile; [2]Universidad Austral de Chile, Valdivia, Chile; [3]Institut für Neuro- und Verhaltensbiologie, University of Münster, Münster, Germany

**Abstract** Mitochondria generate ATP and building blocks for cell growth and regeneration, using pyruvate as the main substrate. Here we introduce PyronicSF, a user-friendly GFP-based sensor of improved dynamic range that enables real-time subcellular quantitation of mitochondrial pyruvate transport, concentration and flux. We report that cultured mouse astrocytes maintain mitochondrial pyruvate in the low micromolar range, below cytosolic pyruvate, which means that the mitochondrial pyruvate carrier MPC is poised to exert ultrasensitive control on the balance between respiration and anaplerosis/gluconeogenesis. The functionality of the sensor in living tissue is demonstrated in the brain of *Drosophila melanogaster* larvae. Mitochondrial subpopulations are known to coexist within a given cell, which differ in their morphology, mobility, membrane potential, and vicinity to other organelles. The present tool can be used to investigate how mitochondrial diversity relates to metabolism, to study the role of MPC in disease, and to screen for small-molecule MPC modulators.

*For correspondence:
fbarros@cecs.cl (LFB);
aalejo@cecs.cl (ASM)

## Introduction

Mitochondria are the chief energy generators of animal cells, accounting for over 90% of ATP production, and they also generate building blocks for the synthesis of sugars, amino acids, nucleic acids and prosthetic groups, essential elements for tissue growth, plasticity and regeneration. In addition to their metabolic functions, mitochondria are involved in diverse physiological and pathophysiological processes, including $Ca^{2+}$ signaling, the production of reactive oxygen species, aging and degeneration, cell death and oncogenesis. Some open issues in mitochondrial physiology are the regulation of intermediate metabolism, the coordination between cytosolic and mitochondrial pathways, the decision between catabolism and anabolism, and the crosstalk between mitochondria and other organelles like plasma membrane and endoplasmic reticulum. Are mitochondria within a given cell metabolically diverse? A major mitochondrial substrate for mammalian cells is pyruvate, a 3-carbon organic acid produced from glucose, lactate and amino acids. Extracellular pyruvate may also reach the cytosol via monocarboxylate transporters (MCTs) (*Halestrap and Price, 1999*). Pyruvate enters mitochondria through the mitochondrial pyruvate carrier (MPC) (*Halestrap and Denton, 1975*; *Herzig et al., 2012*; *Bricker et al., 2012*). Once in the matrix, the flux of pyruvate is split. Carbon is either shed to generate ATP via oxidative phosphorylation or alternatively, carbon is accrued to generate oxaloacetate, a process termed anaplerosis, which provides building blocks for biosynthesis and constitutes the first step of gluconeogenesis.

Pyronic, the first genetically-encoded sensor for pyruvate (*San Martín et al., 2014a*), has permitted the measurement of cytosolic pyruvate in several organisms with high temporal resolution, see

for example (*Mächler et al., 2016*; *Compan et al., 2015*; *Plaçais et al., 2017*; *Delgado et al., 2018*; *Baeza-Lehnert et al., 2019*; *Hasel et al., 2017*; *Rusu et al., 2017*). This article introduces a new version of Pyronic that improves on the original sensor in terms of dynamic range and facility of use, as it is imaged with the 488 nm Argon laser of standard confocal microscopes. To demonstrate the usefulness of the probe, we measured cytosolic, nuclear and mitochondrial pyruvate concentration, MPC-mediated permeability and determined the metabolic flux of small groups of mitochondria. The main finding of this study is that in cultured astrocytes mitochondrial pyruvate lies in the low micromolar range, endowing the MPC with the potential capability of ultrasensitive modulation of anaplerosis. Experiments were also carried out in fruit fly larvae to demonstrate pyruvate dynamics in living tissue.

## Results

### A highly responsive pyruvate sensor

*Figure 1A* illustrates PyronicSF (Single Fluorophore), in which the complete sequence of the bacterial transcription factor PdhR (*Quail and Guest, 1995*) has been linked to a circularly-permuted version of GFP (cpGFP) (*Nagai et al., 2001*) (DNA sequence in *Figure 1—figure supplement 1*). Exposure of the sensor to pyruvate in vitro caused a strong increase in fluorescence when excited by blue light (*Figure 1B*). Excited at 488 nm, the increase in fluorescence emission was ≈ 250%, with a $K_D$ of 480 μM (*Figure 1C*). As expected from the behavior of PdhR (*San Martín et al., 2014a*) in the FRET sensor Pyronic, PyronicSF was insensitive to lactate, to other structurally related organic acids, and to $NAD^+$ and NADH (*Figure 1D*). While the sensing of pyruvate by PyronicSF was not affected by pH, there was a pH-dependent shift of the dose-response curve (*Figure 1—figure supplement 2*). Because the fluorescence ratio of the sensor Pyronic is insensitive to pH in the physiological range (*San Martín et al., 2014a*), we attribute the effect of pH on PyronicSF to the pH-sensitivity typical of cpGFPs (*Marvin et al., 2013*; *Cambronne et al., 2016*). When required, the signal can be corrected by taking advantage of the pH-sensitivity of PyronicSF excited at its isosbestic point (435 nm; *Figure 1B*), as detailed in *Figure 1—figure supplement 2*, or by parallel measurements with a pH probe. Alternatively, experiments may be controlled with Dead-PyronicSF, a mutated version of the sensor almost devoid of pyruvate sensitivity in the physiological range, but as sensitive to pH as PyronicSF (*Figure 1—figure supplement 2*). Excited with violet light, pyruvate caused a decrease in fluorescence emission (*Figure 1B*), which may be exploited to obtain ratiometric measurements using, for example, the 405 nm laser line available in some confocal microscopes. Expressed in the cytosol of HEK293 cells, PyronicSF showed a homogeneous distribution and responded to a saturating pyruvate load with a change in fluorescence similar to that of the purified protein (*Figure 1E*). The amplitude of the response of PyronicSF was >6 times that of Pyronic (*Figure 1E*; *San Martín et al., 2014a*).

### Estimation of MPC activity

PyronicSF was targeted to the mitochondrial matrix using the destination sequence of cytochrome oxidase. The targeting was successful in various cell types, as evidenced by the labeling of elongated cytoplasmic structures that are of shape and size characteristic of mitochondria (*Figure 2—figure supplement 1*). To test the functionality of the sensor we chose astrocytes, cells that combine oxidative phosphorylation with anaplerosis, and that in culture are very thin, ideal to resolve mitochondria (*Figure 2A*). Confirmation of correct targeting was provided by colocalization with the voltage-sensitive dye TMRM (*Figure 2—figure supplement 2*). Mito-PyronicSF also colocalized with the red fluorescent protein mCherry targeted to mitochondria (*Figure 2A* and *Figure 2—figure supplement 1*). Exposure of astrocytes co-expressing mito-PyronicSF and mito-mCherry to pyruvate resulted in a reversible increase in fluorescence ratio, evidencing mitochondrial uptake (*Figure 2B*). The rise in mitochondria was much slower than its accumulation in the cytosol, monitored indirectly with PyronicSF targeted to the nucleus (*Figure 2—figure supplement 3*). In cells in which mito-PyronicSF was expressed by lipid transfection (as opposed to adenoviral transduction), there was sometimes a diffuse staining explained by inefficient mitochondrial targeting/retention. In such 'leaky' cells, the increase in signal elicited by a pyruvate load was biphasic, with a rapid phase attributable to the cytosolic sensor, followed by a slow mitochondrial phase (*Figure 2—figure supplement 3*). Absence

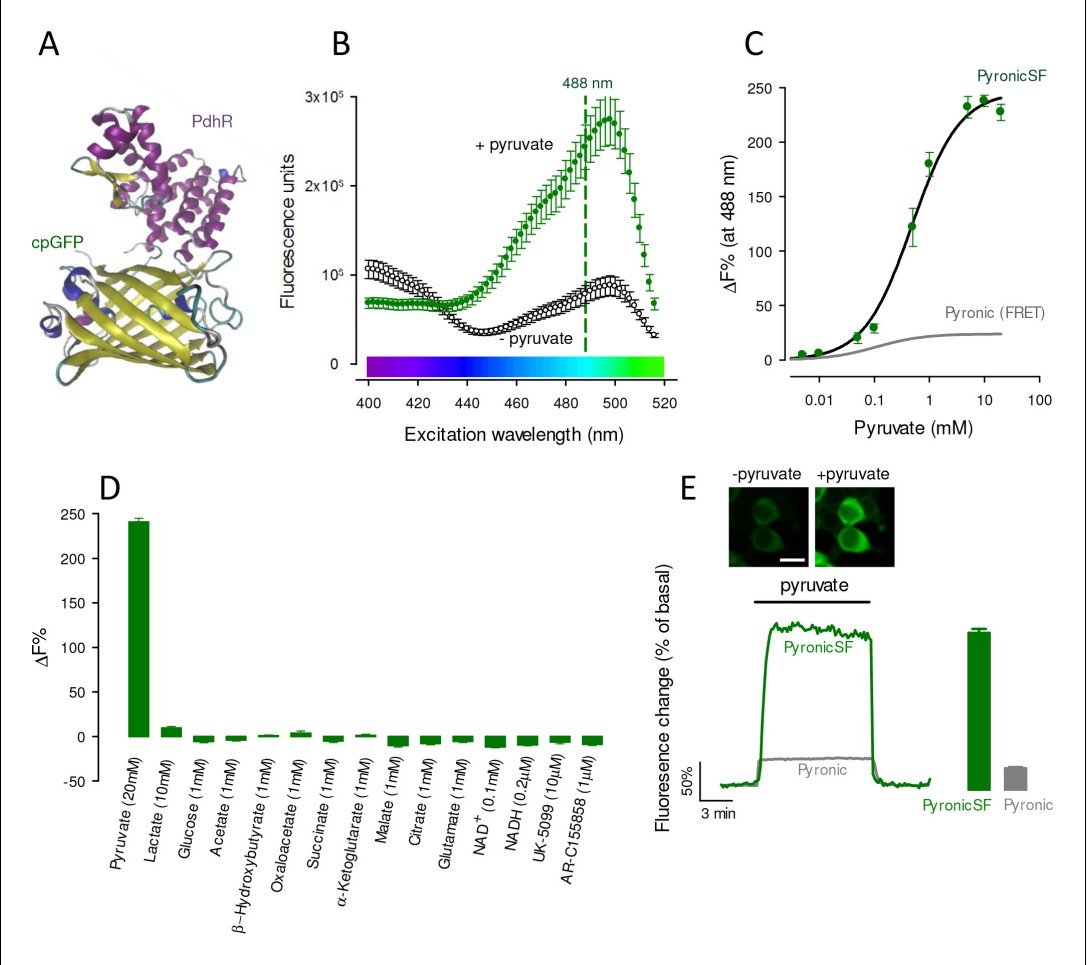

**Figure 1.** Characterization of PyronicSF. (**A**) PyronicSF. cpGFP flanked by linkers was inserted between amino acid residues 188 and 189 of PdhR. DNA sequence in *Figure 1—figure supplement 1*). (**B**) Excitation spectra of PyronicSF in the absence and presence of 10 mM pyruvate. Data are mean ± s.e.m. from 3 protein extracts. (**C**) PyronicSF emission (488 nm excitation) as a function of pyruvate concentration. Data are mean ± s.e.m. from 3 protein extracts. The best fit of a rectangular hyperbola to the data is shown, $K_D = 480 \pm 65$ µM, maximum fluorescence change was 247%. The in vitro saturation curve of the FRET sensor Pyronic is plotted in gray (*San Martín et al., 2014a*). (**D**) PyronicSF emission in the presence of metabolites and transport inhibitors. Data are mean ± s.e.m. from 3 protein extracts. (**E**) Pyruvate dynamics in mammalian cells. HEK293 cells expressing PyronicSF were exposed to 10 mM pyruvate. Images show cells before and during exposure to pyruvate. Bar represents 20 µm. Bar graphs summarize data (mean ± s.e.m.) from 54 cells in four experiments (PyronicSF), and 59 cells in five experiments (Pyronic).

The online version of this article includes the following source data and figure supplement(s) for figure 1:

**Source data 1.** Characterization of PyronicSF.

**Figure supplement 1.** Nucleotide sequence of PyronicSF.

**Figure supplement 2.** Correction of the effect of pH on PyronicSF and a mutant of PyronicSF with reduced response to pyruvate but conserved response to pH.

**Figure supplement 2—source data 1.** Correction of the effect of pH on PyronicSF and a mutant of PyronicSF with reduced response to pyruvate but conserved response to pH.

---

of a rapid phase in the increase of PyronicSF signal in response to extracellular pyruvate suggests correct localization of the sensor in the mitochondrial matrix, as opposed to the intermembrane space. The permeability of the outer mitochondrial membrane, mediated by VDAC, is much higher than that of the inner mitochondrial membrane, mediated by the MPC. Thus, PyronicSF mistargeted to the intermembrane space would have been expected to respond with a faster fluorescence increase, that is, similar to what was observed for the cytosolic sensor. The FRET sensor

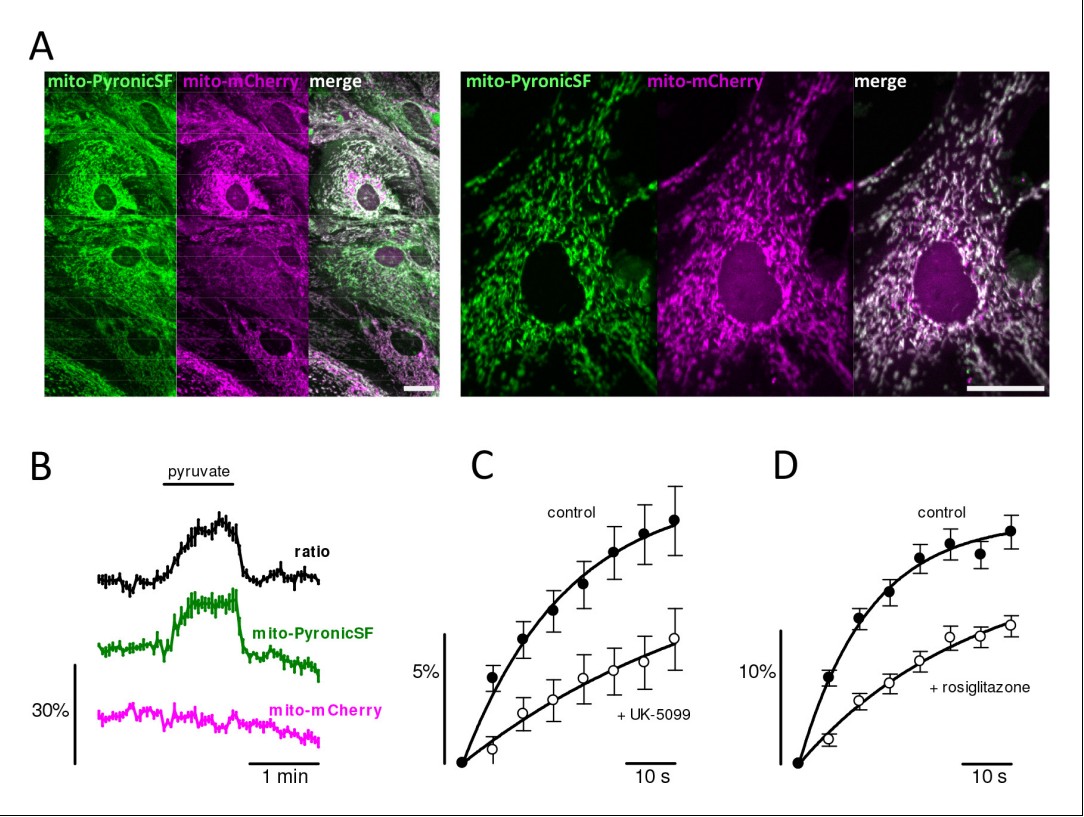

**Figure 2.** MPC-mediated mitochondrial pyruvate transport in astrocytes. (**A**) Astrocytes co-expressing mito-PyronicSF (green) and mito-mCherry (magenta). Bars represent 10 μm. (**B**) Cultures were exposed to 3 mM pyruvate. Data correspond to mean ± s.e.m. (4 cells in a representative experiment). (**C**) Cultures were exposed to 3 mM pyruvate in the absence (black symbols) and presence of 10 μM UK-5099 (white symbols). Data are mean ± s.e.m. of 31 cells from eight experiments. Initial rates (%/min), estimated by fitting a single exponential function to the data (continuous lines), were 32 ± 4 (control) and 10 ± 5 (UK-5099). (**D**) Cultures were exposed to 3 mM pyruvate in the absence (black symbols) and presence of 30 μM rosiglitazone (white symbols). Data are mean ± s.e.m. of 51 cells from nine experiments. Initial rates (%/min), estimated by fitting a single exponential function to the data (continuous lines), were 78 ± 6 (control) and 26 ± 3 (rosiglitazone).

The online version of this article includes the following source data and figure supplement(s) for figure 2:

**Source data 1.** MPC-mediated mitochondrial pyruvate transport in astrocytes.

**Figure supplement 1.** Expression of mito-PyronicSF and mito-Pyronic in various cell types.

**Figure supplement 2.** Mitochondrial localization of mito-PyronicSF in astrocytes.

**Figure supplement 3.** Mitochondrial pyruvate uptake is slower than cytosolic pyruvate uptake.

**Figure supplement 3—source data 1.** Mitochondrial pyruvate uptake is slower than cytosolic pyruvate uptake.

Pyronic could also be targeted to mitochondria in functional form (*Figure 2—figure supplement 1* and *Figure 3—figure supplement 1*), but four copies of the destination sequence were needed, probably because of its larger size relative to PyronicSF.

The uptake of pyruvate by mitochondria was sensitive to the specific MPC1 blocker UK-5099 (*Halestrap and Denton, 1975*; *Herzig et al., 2012*; *Bricker et al., 2012*; *Nagampalli et al., 2018*; *Figure 2C*). The average degree of uptake inhibition was 69% (*Figure 2C*). Possible explanations for the incomplete inhibition may be variations in the subunit composition of the MPC (*Nagampalli et al., 2018*) and/or the presence of alternative routes. This type of uptake protocol seems amenable to the identification and characterization of pharmacological inhibitors of the MPC, a transporter that has been singled out as a target of clinical interest (*Compan et al., 2015*; *Divakaruni et al., 2013*; *Schell et al., 2014*; *Vacanti et al., 2014*; *Gray et al., 2015*; *Olson et al., 2016*; *Divakaruni et al., 2017*; *Li et al., 2017*). To illustrate the potential of mito-PyronicSF for drug discovery, we show here that the insulin sensitizer rosiglitazone inhibited the uptake of pyruvate by

astrocytic mitochondria by 67% (*Figure 2D*), providing confirmation that MPC is a target of thiazoli-dinediones (*Divakaruni et al., 2013*).

## Quantification of mitochondrial pyruvate concentration

The concentration of pyruvate in the mitochondrial matrix of intact cells is unknown. Given the 1:1 stoichiometry between proton and pyruvate transport by the MPC, and the respective pH gradient between cytosol and mitochondria (7.2 and 7.8 [*Azarias et al., 2011*], equivalent to a 4-fold proton gradient), mitochondrial pyruvate may reside anywhere between a few micromolar and four times the cytosolic concentration of pyruvate, a quantity that is neither known with precision. At extracellular levels of glucose, lactate and pyruvate found in brain tissue, steady-state mitochondrial pyruvate was found to lie at the low end of the detection range of PyronicSF (*Figure 3A and C*), which is in agreement with parallel measurements using the FRET sensor (*Figure 3B and C*). A one-point calibration of both sensors was accomplished by forcing a nominal zero cytosolic pyruvate by MCT-accelerated exchange with lactate (*Halestrap and Price, 1999*; *San Martín et al., 2014a*), followed by interpolation in the dose-response curve obtained in vitro (*Figure 1C*). Using this protocol, median mitochondrial pyruvate measured with PyronicSF was 21 µM. Experiments with the sensor targeted to the cytosol showed that the median steady-state concentration of pyruvate in this compartment was 33 µM. The concentration of pyruvate in both compartments varied between cells, explained by differences between experiments (i.e. culture conditions), and also by heterogeneity within a given microscopic field (*Figure 3—figure supplement 2*). We do not know what causes such variability, analogous to that reported in cultured astrocytes for glucose and lactate concentrations, glucose transport, glycolytic rate, and NADH/NAD$^+$ (*San Martín et al., 2013*; *Loaiza et al., 2003*; *Bittner et al., 2010*; *Köhler et al., 2018*). Factors to be explored, ideally with multiple sensors co-expressed, include cell cycle and contact inhibition. There was no apparent correlation between sensor expression level and mitochondrial pyruvate concentration (*Figure 3—figure supplement 3*). To gauge the performance of PyronicSF in the low micromolar range, purified PyronicSF was exposed to low pyruvate concentrations. The experiment was done in vitro to prevent metabolic regulation. *Figure 3—figure supplement 4* shows that the sensor is capable of detecting a difference of 10 µM in the low micromolar range, provided that sufficient replicates are available. This may be a limitation for fast time courses. Estimated with the FRET sensor, whose $K_D$ is five times lower, median pyruvate concentrations were 25 µM in mitochondria and 40 µM in cytosol, a difference of 15 µM. Thus, both sensors converged in showing that mitochondrial pyruvate is in the low micromolar range, and that is lower than cytosolic pyruvate.

## Astrocytic mitochondrial pyruvate is at optimal level for MPC modulation of anaplerosis

With knowledge of pyruvate levels inside and outside mitochondria, a mathematical model could be built to appraise the role of the MPC on catabolism and anabolism. The two main mitochondrial pyruvate sinks are pyruvate dehydrogenase (PDH; EC 1.2.4.1), which is in charge of pyruvate catabolism towards energy production, and pyruvate carboxylase (PC; EC 6.4.1.1), which catalyzes pyruvate anabolism towards anaplerosis and synthesis (*Figure 3D*). Their combined flux was set at 1.2 µM/s (*Fernández-Moncada and Barros, 2014*), with PDH and PC respectively accounting for 76% and 24% (*Oz et al., 2004*). Next, the dosage of MPC was tuned so that at a cytosolic pyruvate of 33 µM and the pH gradient estimated previously in these cells (*Azarias et al., 2011*), mitochondrial pyruvate stabilized at 21 µM, as determined above. With the system configured in this way a number of observations could be made. Firstly, activation of the MPC is necessary for efficient stimulation of oxidative metabolism. For example, PDH activation does not translate into a sustained flux increase because mitochondrial pyruvate becomes depleted, leading to partial PDH desaturation (*Figure 3E*). More dramatically, the fall of pyruvate leads to a strong decrease of flux through PC, a 'steal' flux phenomenon. However, if the activation of PDH is accompanied by activation of the MPC, the flux through PDH can be sustained, with minimal disturbance of PC flux. *Figure 3E* also shows that MPC modulation is sufficient to efficiently modulate the flux through PC. Actually, *Figure 3F* shows that if the steady-state concentration of mitochondrial pyruvate is below 30 µM, a given change in MPC activity will be amplified in terms of PC flux, a phenomenon that has been termed ultrasensitity (*Ferrell and Ha, 2014*). For example, at 21 µM mitochondrial pyruvate, a 1%

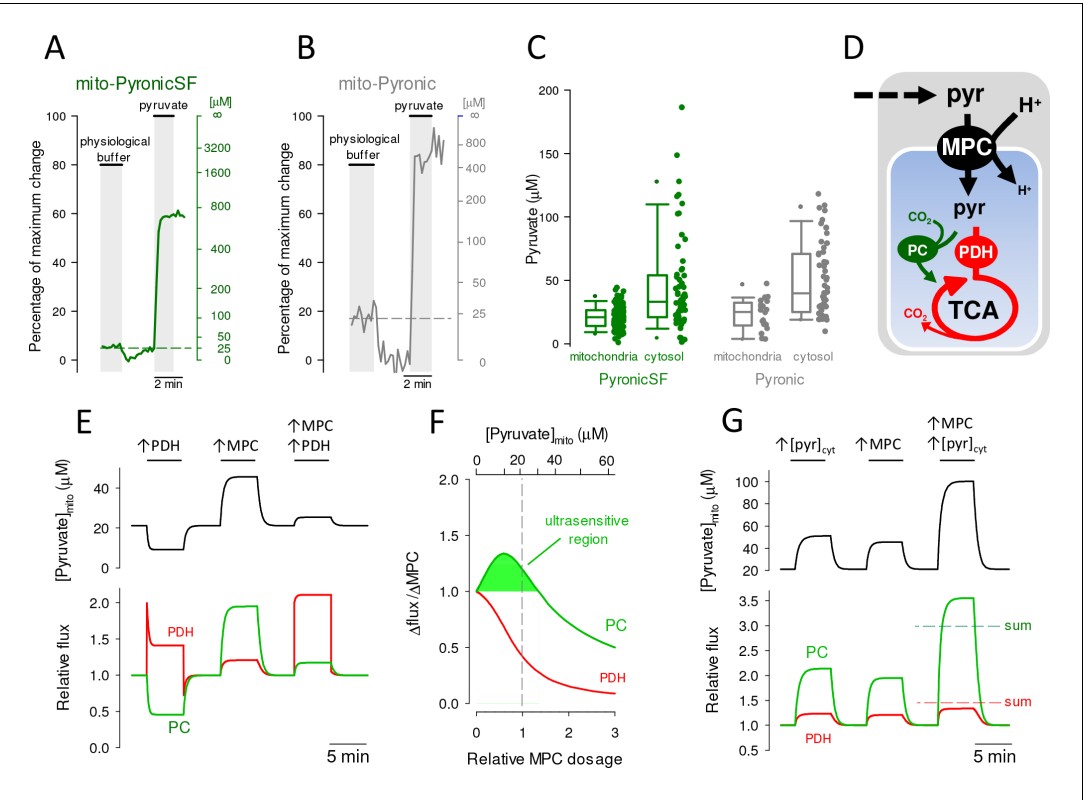

**Figure 3.** Steady-state mitochondrial and cytosolic pyruvate. Astrocytes expressing PyronicSF or Pyronic in mitochondria or cytosol were first incubated in a buffer containing physiological concentrations of glucose (2 mM), lactate (2 mM) and pyruvate (0.2 mM), followed by removal of pyruvate by accelerated-exchange with 10 mM lactate and exposure to 10 mM pyruvate. (**A**) Representative trace from a single astrocyte expressing mito-PyronicSF. Data are shown as percentage of the maximum change (left) and pyruvate concentration (right), with reference to the response of the sensor obtained in vitro (**Figure 1C**). (**B**) Representative trace from a single astrocyte expressing mito-Pyronic. Data are shown as percentage of the maximum change (left) and pyruvate concentration (right), with reference to the response of the sensor obtained in vitro (**San Martín et al., 2014a**). (**C**) Steady-state mitochondrial and cytosolic pyruvate concentrations measured with PyronicSF or Pyronic at physiological concentrations of glucose, lactate and pyruvate, as illustrated in panels A and B. Data are from 131 cells in ten experiments (mito-PyronicSF), 59 cells in four experiments (cytosolic PyronicSF), 20 cells in seven experiments (mito-Pyronic), and 50 cells in five experiments (cytosolic Pyronic). (**D**) Mitochondrial pyruvate dynamics. Pyruvate enters mitochondria (blue compartment) and is metabolized by PDH and the tricarboxylic acid cycle (TCA), or is carboxylated by PC. (**E**) Simulation of pyruvate dynamics in response to PDH and MPC modulation. The effects of activating PDH and MPC by 100% on mitochondrial pyruvate concentration (top panel) and on the fluxes of PDH and PC (bottom panel) are shown. Steady-state cytosolic and mitochondrial pyruvate were 33 μM and 21 μM. Cytosolic and mitochondrial pH were 7.2 and 7.8. Steady-state PDH and PC fluxes were 0.91 and 0.29 μM/s. (**F**) Ultrasensitive modulation of PC flux by the MPC. The curves show the degree of flux increase at PC and PDH relative to the degree of MPC activation. The shaded area under the PC curve indicates the range of pyruvate concentrations at which PC flux increases more than 1% when MPC is activated by 1%. MPC dosage was normalized at 3.24 μM. (**G**) Synergic effect of cytosolic pyruvate and MPC activity on PC flux. The effects of increasing cytosolic pyruvate and MPC activity by 100% on mitochondrial pyruvate concentration (top panel) and on the fluxes of PDH and PC (bottom panel) are shown. The sums of the independent effects are indicated by interrupted lines.

The online version of this article includes the following source data and figure supplement(s) for figure 3:

**Source data 1.** Steady-state mitochondrial and cytosolic pyruvate.
**Figure supplement 1.** Functional expression of the FRET sensor Pyronic in mitochondria.
**Figure supplement 1—source data 1.** Functional expression of the FRET sensor Pyronic in mitochondria.
**Figure supplement 2.** Intra- and inter-experimental contributions to cell-to-cell metabolic heterogeneity of astrocytes.

*Figure 3 continued on next page*

*Figure 3 continued*

**Figure supplement 2—source data 1.** Intra- and inter-experimental contributions to cell-to-cell metabolic heterogeneity of astrocytes.
**Figure supplement 3.** No apparent correlation between PyronicSF expression and mitochondrial pyruvate concentration and consumption.
**Figure supplement 3—source data 1.** No apparent correlation between PyronicSF expression and mitochondrial pyruvate concentration and consumption.
**Figure supplement 4.** Performance of PyronicSF in the low micromolar range.
**Figure supplement 4—source data 1.** Performance of PyronicSF in the low micromolar range.
**Figure supplement 5.** Sensitivity of PyronicSF and Pyronic.
**Figure supplement 5—source data 1.** Sensitivity of PyronicSF and Pyronic.

MPC increase causes a 1.2% increase in PC flux. In contrast, PDH flux increases by only 0.4% (*Figure 3F*). A similar analysis showed that PC flux is also highly responsive to cytosolic pyruvate. For example a rise in cytosolic pyruvate, as may result from glycolytic activation, is faithfully followed by an increase in PC flux and a minor increase in PDH flux (*Figure 3G*). Remarkably, the MPC and cytosolic pyruvate interact synergistically with respect to PC flux but antagonistically with respect to PDH flux. A coincidental rise of MPC activity and cytosolic pyruvate will cause an increase in PC flux larger than the sum of the two independent effects, but for PDH the effects are not even additive (*Figure 3G*). The exquisite sensitivity of PC flux to MPC activity (and to cytosolic pyruvate) is not observed at high mitochondrial pyruvate levels (*Figure 3F*). Together, these results indicate that the MPC is poised for sensitive modulation of anaplerosis. This conclusion stems from the current finding of low relative intramitochondrial pyruvate and the assumption that in astrocytes PC has significantly lower affinity for pyruvate than PDH, as observed in liver and cell lines (BRENDA database). Of note, the behavior of the system is essentially insensitive to the specific kinetic parameters of the MPC (data not shown), as expected for a transporter working far from thermodynamic equilibrium.

## Quantification of mitochondrial pyruvate flux

A protocol to approach flux was devised based on the idea that acute inhibition of mitochondrial pyruvate entry at the MPC should cause a progressive fall in mitochondrial pyruvate as it is consumed by PDH and PC. Analogous methods have been applied successfully to the measurement of whole cell glucose, lactate and pyruvate consumptions in various cell types (*Baeza-Lehnert et al., 2019*; *San Martín et al., 2014b*; *Barros, 2018*). As shown in *Figure 4A*, MPC stoppage with UK-5099 caused immediate depletion of mitochondrial pyruvate. Consistent with the partial UK-5099 inhibition of the pyruvate transport described above, the depletion was not complete. We do not think that the partial effect is explained by insufficient UK-5099, because similar steady-states were reached at 0.5 μM and 10 μM UK-5099 (53 ± 12% versus 47 ± 6%; 12 cells in three paired experiments, p>0.05, data not shown) and the half-inhibition constant for MPC inhibition by UK-5099 is only 50 nM (*Halestrap, 1975*). Thus, the present protocol underestimates the actual rate of mitochondrial pyruvate consumption by an approximate factor of 2. Reportedly, long term exposure to UK-5099 causes irreversible inhibition of the MPC (*Divakaruni et al., 2013*; *Hildyard et al., 2005*) but this was not observed at the low concentrations and short exposure times used here (*Figure 4B*). This reversibility opens the possibility of before-and-after experiments. Using such a protocol, uncoupling of oxidative phosphorylation (OXPHOS) strongly stimulated mitochondrial pyruvate consumption (*Figure 4C*) whereas conversely, inhibition of OXPHOS with the cytochrome oxidase inhibitor azide resulted in a lower rate (*Figure 4D*). A representative example of the estimation of mitochondrial pyruvate consumption expressed in absolute terms is shown in *Figure 4E*. There was substantial variability between cells, which ranged from 0.29 to 3.1 μM/s, with a median value of 1.1 μM/s (*Figure 4E*). There was no apparent correlation between sensor expression level and the rate of mitochondrial pyruvate consumption (*Figure 3—figure supplement 3*).

To exemplify how mito-PyronicSF might serve to study subcellular metabolism, pyruvate concentration and consumption were determined in mitochondria located in different regions of a cell. Steady-state pyruvate in a small group of peripheral mitochondria was found to be higher than in mitochondria lying close to the nucleus. Application of the transporter-block protocol showed that pyruvate consumption may also differ within a cell (*Figure 5*).

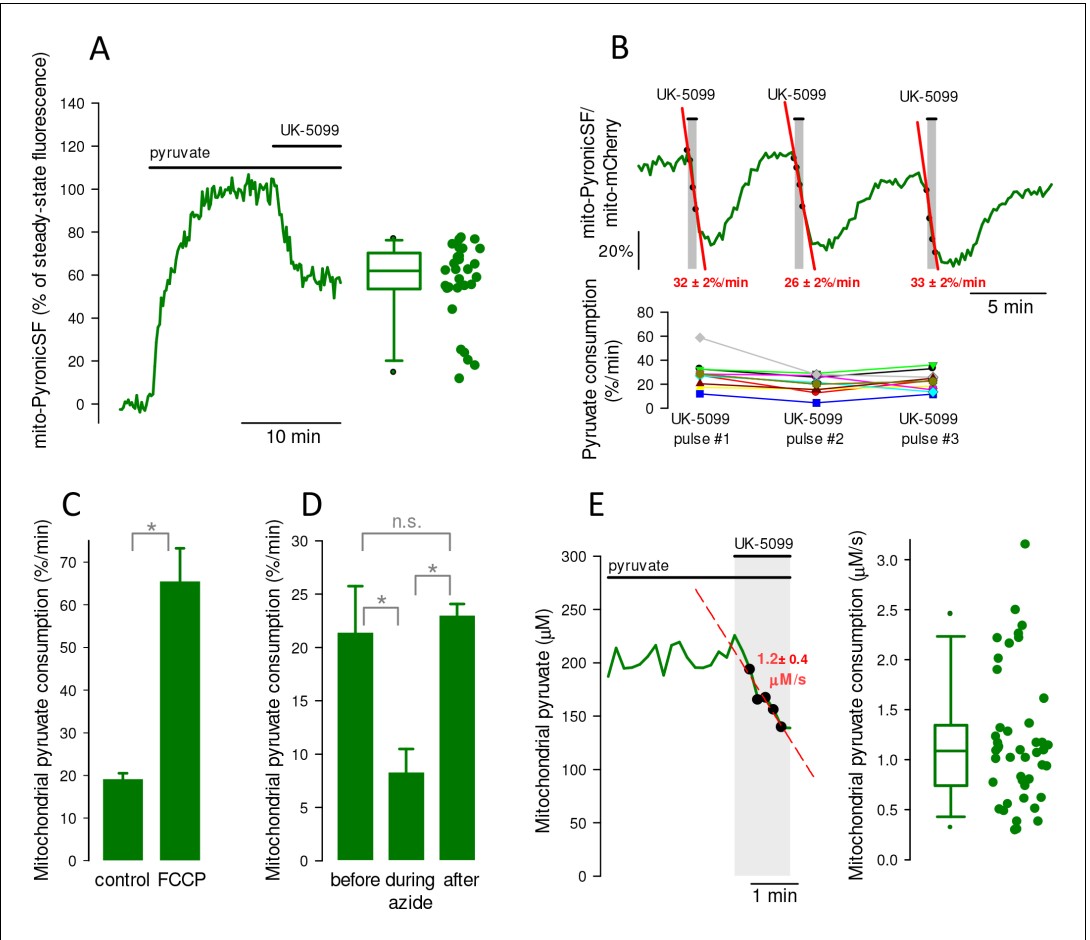

**Figure 4.** Measurement of mitochondrial pyruvate consumption rate in individual cells. (**A**) Time course of mitochondrial pyruvate level after an astrocyte was exposed 5 mM pyruvate and then to 10 μM UK-5099. The new steady-state is represented on the right, as percentage of the level before addition of UK-5099. Median = 62%. Data are from 29 cells in ten experiments. (**B**) An astrocyte incubated in 5 mM pyruvate was exposed three times for 30 s to 0.5 μM UK-5099. Rates of pyruvate depletion are shown in red. The result of three similar experiments (9 cells) is shown below. (**C**) The rate of mitochondrial pyruvate depletion induced with 10 μM UK-5099 was monitored before and after exposure to the proton ionophore FCCP (1 μM). Data are mean ± s.e.m (29 cells in three experiments). (**D**) The rate of mitochondrial pyruvate depletion induced with 10 μM UK-5099 was monitored before, during and after exposure to the cytochrome oxidase inhibitor azide (5 mM; 16 cells in three experiments). (**E**) An astrocyte superfused with 3 mM pyruvate was exposed to 10 μM UK-5099, resulting in a rate of depletion of 1.2 μM/s. The right panel represents the summary of sixteen experiments (44 cells).

The online version of this article includes the following source data for figure 4:

**Source data 1.** Measurement of mitochondrial pyruvate consumption rate in individual cells.

## Mitochondrial pyruvate dynamics in brain tissue

The ability of mito-PyronicSF to monitor pyruvate in living tissue was investigated in *Drosophila mel-anogaster* larvae. For facility of access, we studied perineurial glial cells, which form a monolayer separating the brain from the surrounding hemolymph. PyronicSF expressed very well in cytosol and mitochondria of these cells (*Figure 6A–B*). Superfusion of acutely isolated brains with pyruvate resulted in a quick increase in cytosolic pyruvate, revealing the presence of abundant surface pyru-vate transporters in these cells (*Figure 6C–E*). The response of mitochondria was slower and pla-teaued at lower pyruvate levels, consistent with mitochondria being a site of pyruvate consumption downstream of the cytosol (*Figure 6D–E*). In the presence of a buffer containing glucose, lactate and pyruvate, the steady-state level of pyruvate was much higher in the cytosol than in mitochondria (*Figure 6C–D*). Experiments are planned to measure transmitochondrial pyruvate and pH gradients

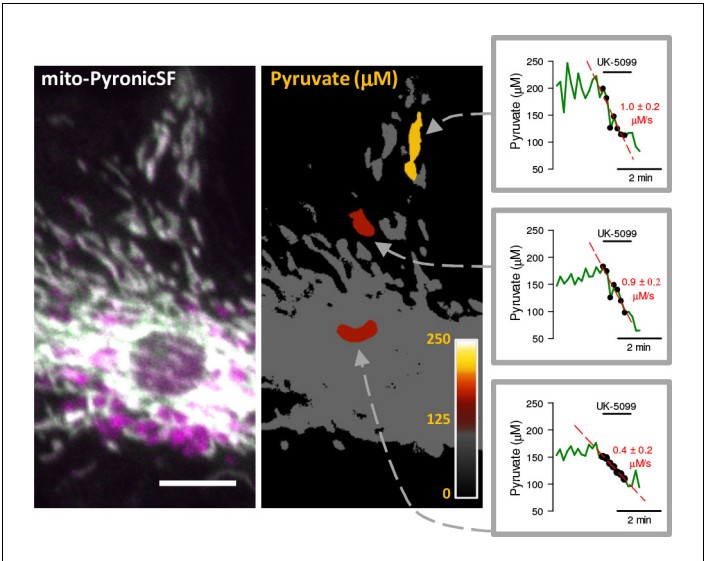

**Figure 5.** Pyruvate concentration and consumption in discrete mitochondria. Mito-PyronicSF (gray) in an astrocyte expressing mito-PyronicSF and mito-mCherry (left). Bar represents 10 µm. The righthand image shows three regions of interest colored according to the look-up table in the inset (0 to 250 µM pyruvate, calibrated as described for *Figure 3*). The three graphs show pyruvate consumption rates in the regions of interest, determined with 10 µM UK-5099 in cells incubated with 3 mM pyruvate.

The online version of this article includes the following source data for figure 5:

**Source data 1.** Pyruvate concentration and consumption in discrete mitochondria.

in the presence of normal hemolymph substrates. Nevertheless, the steep transmitochondrial pyruvate gradient measured here suggests that the MPC is also a key regulator of the balance between catabolism and anabolism in perineurial glial cells.

## Discussion

The present article introduces PyronicSF, a new tool for the study of metabolism. PyronicSF is genetically-encoded and compatible with standard fluorescence microscopes. In combination with *ad-hoc* protocols, this sensor permits the measurement of transport, concentration and flux of pyruvate in intact mitochondria. In combination with suitable experimental models, PyronicSF may be adapted to the analysis of intact organs, cell populations, single cells or even individual mitochondria. Demonstrating its potential, we showed that in mouse astrocytes and probably in perineurial cells from *Drosophila melanogaster*, mitochondrial pyruvate lies in the low micromolar range, a finding that informs about the regulatory role of the MPC.

### A new approach to mitochondrial function

Mitochondria are the chief ATP producers of eukaryotic cells and also a principal site for the generation of building blocks for macromolecular synthesis. The speeds of these catabolic and anabolic pathways are assessed by diverse techniques of complementary strengths and weaknesses (*Barros, 2018*). For catabolism, respirometry is the standard technique since the days of Otto Warburg. The rate of oxygen consumption, which is tightly linked to the production of ATP, is specific and quantitative but requires cell populations, typically more than 10,000 cells in modern devices. Isotopic techniques were introduced later on to map specific metabolic pathways, although with low temporal resolution. They also require many cells. These direct measures of metabolic flux have been complemented by fluorescent indicators of pH, membrane potential, free radicals and calcium. Thanks to the spatiotemporal resolution of fluorescence microscopy they afford measurements in the range of seconds and the ability to resolve up to single organelles, but are not informative about the speed of metabolism (*Brand and Nicholls, 2011*). A few years ago, our group introduced the FRET sensor Pyronic (*San Martín et al., 2014a*) that has permitted high resolution measurement of

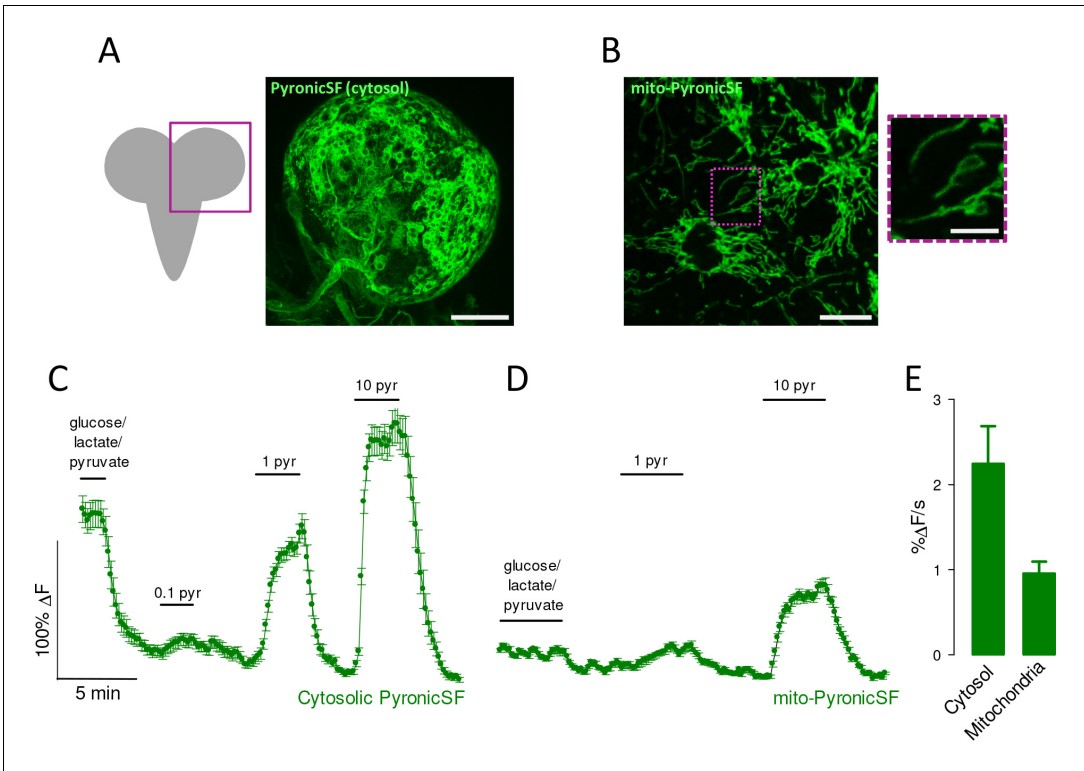

**Figure 6.** Pyruvate dynamics in glial cells of *Drosophila melanogaster*. Brains were acutely dissected from *Drosophila melanogaster* larvae expressing PyronicSF in the cytosol or mitochondria of perineurial glial cells. (**A**) PyronicSF in the cytosol of perineurial cells. Bar represents 100 μm. (**B**) Mito-PyronicSF in perineurial cells. Bar represents 10 μm. An area containing clearly identifiable mitochondria is shown under higher magnification on the right. Bar represents 5 μm. (**C**) A brain expressing cytosolic PyronicSF in perineurial cells was superfused with HL3 buffer containing 5 mM glucose, 1 mM lactate and 0.5 mM pyruvate. After removal of the substrates, the tissue was sequentially exposed to 0.1, 1 and 10 mM pyruvate. Data are mean ± s.e.m. (20 cells). (**D**) A brain expressing mito-PyronicSF in perineurial cells was superfused with HL3 buffer containing 5 mM glucose, 1 mM lactate and 0.5 mM pyruvate. After removal of the substrates, the tissue was sequentially exposed to 1 and 10 mM pyruvate. Data are mean ± s.e.m. (20 cells). (**E**) Rates of PyronicSF fluorescence increase in response to 10 mM pyruvate. Data are mean ± s.e.m. (60 cells from three experiments similar to those shown in C-D).

The online version of this article includes the following source data for figure 6:

**Source data 1.** Pyruvate dynamics in glial cells of *Drosophila melanogaster*.

pyruvate in multiple contexts from cultured cells to living animals (*Mächler et al., 2016*; *Compan et al., 2015*; *Plaçais et al., 2017*; *Delgado et al., 2018*; *Baeza-Lehnert et al., 2019*; *Hasel et al., 2017*; *Rusu et al., 2017*). The present sensor PyronicSF improves on the original sensor on several grounds: i. It has a higher dynamic range. ii. It does not require FRET detection technology, so it can be used in standard wide-field and confocal microscopes. iii. It is smaller and therefore more easily targetable to subcellular domains. iv. Because of its lower affinity for pyruvate, when expressed in mitochondria it provides an extended window for rate monitoring. v. It can estimate mitochondrial pyruvate consumption under physiological conditions (i.e. in the presence of glucose). vi. It permits monitoring pyruvate concentration, permeability and flux in individual mitochondria, thus offering a practical tool for the functional study of mitochondrial heterogeneity. *Figure 3—figure supplement 5* shows that PyronicSF is more sensitive than Pyronic at all pyruvate concentrations. One limitation of PyronicSF is its pH sensitivity, which can be corrected as explained in the Results Section. In principle, buffering by a sensor may interfere with fast ligand dynamics, as happens with $Ca^{2+}$ sensing (*Grienberger and Konnerth, 2012*), but this problem is not anticipated for pyruvate because its levels are higher (μM instead of nM) and metabolic transients are slow (seconds instead of ms). Possible interference of surface permeability in the study of MPC transport activity may be

circumvented using permeabilized cells. Assays based on mito-PyronicSF may be combined with the study of the MPC using luminescence (*Compan et al., 2015*).

## Ultrasensitive regulation of anaplerosis/gluconeogenesis by the MPC

Why is the mitochondrial concentration of pyruvate important? After it became established that pyruvate enters mitochondria through a transporter and not via simple diffusion (*Papa et al., 1971*; *Halestrap and Denton, 1974*), issue was raised as to the role of the transporter on pyruvate utilization (*Shearman and Halestrap, 1984*; *Schell and Rutter, 2013*). This is not a trivial question, as some transporters play important roles in flux control, for example glutamate transporters in the brain, and others do not, such as monocarboxylate carriers in most tissues (*Barros and Deitmer, 2010*). Key to this role is the thermodynamic gradient driving the transporter. For the MPC, the relevant parameters are pH and pyruvate. It is well established that there is steep transmitochondrial proton gradient, which ranges between 0.6 and 1 pH units (*Azarias et al., 2011*; *Nicholls and Ferguson, 2013*), but for pyruvate, the numbers have not been available. Pyruvate in samples of brain tissue obtained by quick-freezing was measured at 99 µM (*Hawkins et al., 1973*), although fast glycogen degradation during the procedure may have affected the result (*Swanson et al., 1992*; *Oe et al., 2016*). The concentrations found here in cultured astrocytes together with the different pyruvate affinity of PC and PDH indicates that the MPC is poised to modulate metabolic decisions. A caveat however is that our mathematical simulation assumed astrocyte values from enzyme saturation parameters obtained in other mammalian cells (BRENDA database). Enzyme determinations in astrocytes are needed to confirm this assumption. Would that be the case in other cell types? Gluconeogenesis from lactate and amino acids in liver feeds the brain with glucose during fasting and exercise, but becomes excessive in diabetes, leading to hyperglycemia and chronic illness. Gluconeogenesis starts with the carboxylation of pyruvate by pyruvate carboxylase (PC) and this step may have a more important role in rate modulation than anticipated (*Gray et al., 2015*; *Burgess et al., 2007*). The MPC has also been pinpointed as a target in neurodegenerative diseases, because its inhibition mobilizes glutamate towards oxidation therefore ameliorating excitotoxicity (*Divakaruni et al., 2017*). In contrast, in cancer cells, MPC acts as a protective factor by inhibiting the Warburg effect (*Schell et al., 2014*) and MPC inhibition enhanced tumor aggression and resistance to chemo- and radiotherapy (*Schell et al., 2014*; *Li et al., 2017*). Growth under MPC-inhibition is sustained by reprogramming of mitochondrial metabolism including enhanced glutaminolysis and lipid catabolism (*Vacanti et al., 2014*). Knowledge of mitochondrial pyruvate concentration, permeability and flux at cellular and subcellular levels may contribute to the understanding of these complex diseases, as well as in the identification and characterization of drugs that target the MPC, such as thiazoledinediones (*Divakaruni et al., 2013*).

# Materials and methods

## Key resources table

| Reagent type (species) or resource | Designation | Source or reference | Identifiers | Additional information |
|---|---|---|---|---|
| strain, strain background (*Mus musculus*) | C57BL/6J x CBA/J | The Jackson Laboratory | JAX: 100011 RRID:IMSR_JAX:100011 | |
| Genetic reagent (*Drosophila melanogaster*) | Bloomington 45781 | Bloomington *Drosophila* Stock Center | apt-Gal4 (P[GMR49G07-GAL4]attP2) RRID:BDSC_45781 | |
| cell line (*Homo sapiens*) | HEK293 | ATCC | CRL-1573 RRID:CVCL_0045 | |
| cell line (*Homo sapiens*) | HeLa | ATCC | CRM-CCL-2 RRID:CVCL_0030 | |

*Continued on next page*

*Continued*

| Reagent type (species) or resource | Designation | Source or reference | Identifiers | Additional information |
|---|---|---|---|---|
| cell line (*Homo sapiens*) | MDA-MB-231 | ATCC | CRM-HTB-26 RRID:CVCL_0062 | |
| cell line (*Cercopithecus aethiops*) | COS7 | ATCC | CRL-1651 RRID:CVCL_0224 | |
| recombinant DNA reagent | Plasmid: Mito-PyronicSF/ pCMV-myc-mito | *Arce-Molina et al., 2019* | RRID:Addgene_124813 | |
| recombinant DNA reagent | Plasmid: NLS-PyronicSF/ pCMV-myc-nuc | *Arce-Molina et al., 2019* | RRID:Addgene_124814 | |
| recombinant DNA reagent | Plasmid: PyronicSF-mRuby2/pBI-CMV1 | *Arce-Molina et al., 2019* | RRID:Addgene_124830 | |
| recombinant DNA reagent | Adenoviral particles: Ad mito-PyronicSF-T2A-mito-mCherry | Vector Biolabs | Custom made | |
| recombinant DNA reagent | Adenoviral particles: Ad Pyronic | Vector Biolabs | Custom made | |
| recombinant DNA reagent | Plasmid: Pyronic | *San Martín et al., 2014a* | RRID:Addgene_51308 | |
| commercial assay or kit | Lipofectamine 3000 Transfection Reagent | Invitrogen | Cat. #L3000015 | |
| chemical compound, drug | UK-5099 | TOCRIS | CAS: 56396-35-1 | |
| chemical compound, drug | AR-C155858 | Haoyuan Chemexpress | CAS: 496791-37-8 | |
| software, algorithm | Berkeley Madonna 8.3.23.0 | University of California at Berkeley | http://www.berkeleymadonna.com | |
| software, algorithm | Fluoview FV10-ASW 3.0 | Olympus | N/A | |
| software, algorithm | ImageJ 1.49 m | NIH | http://imagej.net | |

Standard reagents were acquired from Sigma and Merck.

## Generation and in vitro characterization of PyronicSF

Single fluorophore pyruvate sensors were built fusing PdhR (*Quail and Guest, 1995*) and the circularly-permutted GFP (cpGFP *Nagai et al., 2001*). PdhR is a transcriptional regulator from *Escherichia coli* that controls the expression of PDH (*Quail and Guest, 1995*) and was previously used as a specific pyruvate-binding domain in the FRET sensor Pyronic (*San Martín et al., 2014a*). The fluorophore cpGFP is a variant of GFP in which the two ends of the protein are linked with the hexapeptide GGTGGS, while new ends were generated by cleaving the protein between amino acid residues N144 and Y145 and replacing the latter by a methionine to serve as new N terminus (*Baird et al., 1999*). Using a 3D model structure of PdhR obtained with the M4T server 3.0 from Fiser Laboratory, three sensor variants were designed by inserting cpGFP after amino acid residues 94, 122 and 189 of PdhR, that is, within loops of the ligand binding domain with free access to solvent and far from the hydrophobic core of the protein. To obtain purified protein, the DNA-coding sequences were cloned into pGST-Paralell1 (*Sheffield et al., 1999*) and used to transform

competent *E. coli* BL21 (DE3). Single colonies were grown in LB medium with 100 mg/ml ampicillin and expression was induced with 400 μM IPTG. Cells were collected by centrifugation at 5000 rpm (4°C) for 10 min and disrupted by sonication (Hielscher Ultrasound Technology) in 5 mL of Tris-HCl buffer pH 8.0. Cell-free extracts were obtained by centrifugation at 10,000 rpm (4°C) for 1 hr. Proteins were purified using a glutathione resin as recommended by the manufacturer (GST Buffer and Resin Kit, General Electric Healthcare Life Sciences). After cleavage of GST with protease TEV (AustralProteins), eluted proteins were incubated overnight at 4°C and measured with a microplate reader analyzer (EnVision, PerkinElmer). Excitation spectra were obtained collecting emission at 515 ± 15 nm. The variant with cpGFP inserted after the amino acid residue 189 of PdhR showed the largest change in fluorescence intensity upon binding pyruvate and was christened PyronicSF (PYRuvate Optical Nano-Indicator from CECs Single Fluorophore; DNA sequence is available in *Figure 1—figure supplement 1*). The sequence was cloned into pcDNA3.1(-) for expression in eukaryotic cells. A version of the sensor with greatly reduced sensitivity to pyruvate and conserved sensitivity to pH (Dead-PyronicSF; *Figure 1—figure supplement 2*) was obtained by mutating the first linker CTCGAA (red in *Figure 1—figure supplement 1*, coding for Leu-Glu) to ACCATG (coding for Tre-Met). PyronicSF was targeted to mitochondria or nucleus using pSHOOTER plasmids (ThermoFisher). PyronicSF plasmids are available from Addgene.

## Animals and cultures (mice and flies)

The reports of the procedures comply with the ARRIVE guidelines. Mixed F1 mice (C57BL/6J x CBA/J) were kept in an animal room under Specific Pathogen Free (SPF) conditions at a room temperature of 20 ± 2°C, in a 12/12 hr light/dark cycle with free access to food and water. Procedures were approved by the Centro de Estudios Científicos Animal Care and Use Committee, project 1160317. Mixed cortical cultures (2–3 day-old neonates) were prepared as detailed previously *Bittner et al. (2010)*. Astrocytes in cortical cultures were used at days 8–10. Cell lines were acquired from the American Type Culture Collection (ATCC) and tested free of mycoplasma. Cells were cultured at 37°C in 95% air/5% $CO_2$ in DMEM/F12 (HEK293; CRL-1573), EMEM (HeLa; CRM-CCL-2), DMEM high glucose (COS7; CRL-1651), supplemented with 10% fetal bovine serum. MDA-MB-231 (CRM-HTB-26) were cultured in Leibovitz's L-15 Medium supplemented with 10% fetal bovine serum at 37°C in a free gas exchange incubator (no $CO_2$ injection). Cells were transfected at 60% confluence using Lipofectamine 3000 (Invitrogen) or alternatively, exposed to $5 \times 10^6$ PFU of Ad Pyronic (serotype 5, custom made by Vector Biolabs), and studied after 16–24 hr. Flies were kept at 25°C on a standard diet. The fly stock used in this study was apt-Gal4 (P[GMR49G07-GAL4]attP2, Bloomington 45781). The pyruvate sensor expressing fly stocks UAS-PyronicSF and UAS-Mito-Pyronic were generated using the following strategy. The coding sequence of the sensors was amplified by PCR and cloned into a pENTR-vector using pENTR/D-TOPO Cloning (ThermoFisher). Then, the sensor coding sequences were cloned via gateway cloning (Gateway LR, Invitrogen) into the vector pUASTattBrfa (*Bischof et al., 2013*), which allows Φintegrase-mediated integration into the fly genome. The resulting vectors have been integrated into the fly genome at landing site attP40.

## Imaging

Cultured cells were imaged at room temperature (22–25°C) in KRH buffer of the following composition (in mM): 136 NaCl, 3 KCl, 1.25 $CaCl_2$, 1.25 $MgSO_4$, 1–2 glucose, 10 HEPES pH 7.4, using an upright Olympus FV1000 confocal microscope equipped with 440, 488 nm and 543 nm laser lines and a 20X water immersion objective (NA 1.0). Pyronic was imaged at 440 nm excitation/480 ± 15 nm (mTFP) and 550 ± 15 (Venus) emissions. mCherry and TMRM were imaged at 543 nm excitation/610 ± 50 emission. Third instar larval *Drosophila* brains were dissected in HL3 buffer (70 mM NaCl, 5 mM KCl, 20 mM $MgCl_2$, 10 mM $NaHCO_3$, 115 mM sucrose, 5 mM trehalose, 5 mM HEPES; pH 7.1) and placed into a flow-through imaging chamber (Warner Instruments, Hamden, USA). The chamber was fixed onto the stage of a Leica SP8 microscope (Leica Microsystems, Wetzlar, Germany) and brains were superfused with HL3 buffer. Images were acquired using a 40X oil immersion objective (NA 1.4). PyronicSF was imaged at 488 nm excitation/525 ± 25 emission. Eight-bit images of PyronicSF (GFP and mCherry channels) and Pyronic (mTFP and Venus channels) were processed using ImageJ (Rasband, W.S., ImageJ, U. S. National Institutes of Health, Bethesda, Maryland, USA). The lowest pixel intensity value of each stack was subtracted from every frame, and a binary mask was

generated using the sum of both stacks (Process/Image Calculator/Add). Mask edges were defined with the 'mixture modelling' plugin as the intercept between Gaussian distributions representing mitochondrial signal and background. Regions of interest (ROIs) were selected on the masked stacks.

## Mathematical modeling

Mitochondrial pyruvate dynamics were simulated using Berkeley Madonna software and the following set of ordinary differential equations:

$$dMPC_o/dt = K_{off}H^*MPC_oH + f_1{}^*MPC_i - K_{on}{}^*MPC_o{}^*H_o - f_1{}^*MPC_o \tag{1}$$

$$dMPC_i/dt = K_{off}H^*MPC_iH + f_1{}^*MPC_o - K_{on}{}^*MPC_i{}^*H_i - f_1{}^*MPC_i \tag{2}$$

$$dMPC_oH/dt = K_{on}{}^*MPC_o{}^*H_o + K_{off}P^*MPC_oHP - K_{off}H^*MPC_oH - K_{on}{}^*MPC_oH^*P_o \tag{3}$$

$$dMPC_iH/dt = K_{on}{}^*MPC_i{}^*H_i + K_{off}P^*MPC_iHP - K_{off}H^*MPC_iH - K_{on}{}^*MPC_iH^*P_i \tag{4}$$

$$dMPC_oHP/dt = K_{on}{}^*MPC_oH^*P_o + f_2{}^*MPC_iHP - K_{off}P^*MPC_oHP - f_2{}^*MPC_oHP \tag{5}$$

$$dMPC_iHP/dt = K_{on}{}^*MPC_iH^*P_i + f_2{}^*MPC_oHP - K_{off}P^*MPC_iHP - f_2{}^*MPC_iHP \tag{6}$$

$$dP_i/dt = K_{off}P^*MPC_iHP - K_{on}{}^*MPC_iH^*P_i - PDH - PC \tag{7}$$

*Equations 1 to 6* represent the six conformations of the MPC carrier: outward- and inward-facing, either empty ($MPC_o$ and $MPC_i$), loaded with a proton ($MPC_oH$ and $MPC_iH$) or loaded with both proton and pyruvate ($MPC_oHP$ and $MPC_iHP$). *Equation 7* describes intramitochondrial pyruvate. MPC was set at 3.24 µM; the association constant $K_{on}$ for protons and pyruvate was set at $10^8$ $M^{-1}{*}s^{-1}$; the dissociation constants $K_{off}H$ and $K_{off}P$ were 20 $s^{-1}$ and $2.12{*}10^6$ $s^{-1}$; carrier translocation rates $f_1$ (empty) and $f_2$ (loaded) were set at 200 $s^{-1}$ and 3000 $s^{-1}$. Pyruvate dehydrogenase (PDH) was modeled as a saturable reaction with $V_{max}$ of 1.34 µM* $s^{-1}$ and $K_m$ of 10 µM (BRENDA Enzyme Database). Pyruvate carboxylase (PC) was modeled as a saturable reaction with $V_{max}$ of 3.3 µM* $s^{-1}$ and $K_m$ of 220 µM (BRENDA Enzyme Database). With these parameters and cytosolic and mitochondrial pH values of 7.2 (63 nM) and 7.8 (16 nM *Azarias et al., 2011*), the simulated carrier displayed accelerated exchange and an apparent zero-trans $K_m$ ($K_{zt}$) of 150 µM (*Halestrap, 1975*). Mitochondrial pyruvate ($P_i$) stabilized at 21 µM when cytosolic pyruvate ($P_o$) was 33 µM and mitochondrial pyruvate consumption was 1.2 µM* $s^{-1}$ [30], with PDH and PC respectively accounting for 76% and 24% of the flux (*Oz et al., 2004*).

## Statistical analysis

Statistical analyses were carried out with SigmaPlot software (Jandel). For normally distributed variables, differences were assessed with the Student´s t-test (pairs) and with ANOVA followed by the Tukey-Kramer *ad hoc* test (groups). For variables that failed the normality test, differences were assessed with the Mann-Whitney Rank Sum Test (pairs) or the Kruskal-Wallis one way ANOVA on ranks (groups). *, $p < 0.05$; ns (non-significant), $p > 0.05$. The number of experiments and cells is detailed in each figure.

## Acknowledgements

We thank Karen Everett for critical reading of the manuscript. This work was partly supported by Fondecyt grants 11150930 to ASM and 1160317 to LFB, and Deutsche Forschungsgemeinschaft (DFG) grants SFB1009 and SCHI 1380/2 to SS. The Centro de Estudios Científicos (CECs) is funded by the Chilean Government through the Centers of Excellence Basal Financing Program of CONICYT.

## Additional information

### Funding

| Funder | Grant reference number | Author |
|---|---|---|
| Fondo Nacional de Desarrollo Científico y Tecnológico | 11150930 | Alejandro San Martín |
| Fondo Nacional de Desarrollo Científico y Tecnológico | 1160317 | L Felipe Barros |
| Comisión Nacional de Investigación Científica y Tecnológica | PB-01 | L Felipe Barros |
| Deutsche Forschungsgemeinschaft | SFB1009 | Stefanie Schirmeier |
| Deutsche Forschungsgemeinschaft | SCHI 1380/2 | Stefanie Schirmeier |

The funders had no role in study design, data collection and interpretation, or the decision to submit the work for publication.

### Author contributions

Robinson Arce-Molina, Francisca Cortés-Molina, Pamela Y Sandoval, Alex Galaz, Karin Alegría, Data curation, Formal analysis, Investigation, Methodology, Writing - review and editing; Stefanie Schirmeier, Conceptualization, Resources, Data curation, Formal analysis, Supervision, Funding acquisition, Investigation, Methodology, Project administration, Writing - review and editing; L Felipe Barros, Conceptualization, Data curation, Formal analysis, Supervision, Funding acquisition, Writing - original draft, Project administration, Writing - review and editing; Alejandro San Martín, Conceptualization, Data curation, Formal analysis, Supervision, Funding acquisition, Investigation, Methodology, Writing - original draft, Project administration, Writing - review and editing

### Author ORCIDs

L Felipe Barros  https://orcid.org/0000-0002-6623-4833
Alejandro San Martín  https://orcid.org/0000-0002-5608-5117

### Ethics

Animal experimentation: This study was performed in strict accordance with the recommendations in the Guide for the Care and Use of Laboratory Animals of the National Institutes of Health. Procedures were approved by the Centro de Estudios Científicos Animal Care and Use Committee, project 1160317.

### Decision letter and Author response

Decision letter https://doi.org/10.7554/eLife.53917.sa1
Author response https://doi.org/10.7554/eLife.53917.sa2

## Additional files

### Supplementary files

• Transparent reporting form

### Data availability

All data generated or analysed during this study are included in the manuscript and supporting files. Source data files have been provided for all figures.

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
