## [Decision Letter]

**Acceptance summary:**

You have done an excellent job of addressing the reviewer's comments and it was agreed that your manuscript is much improved. We believe that your sensor will provide a valuable tool for others in studying cellular metabolism and that this manuscript is a major step forward.

**Decision letter after peer review:**

Thank you for resubmitting your work entitled "A highly responsive pyruvate sensor reveals pathway-regulatory role of the mitochondrial pyruvate carrier MPC" for further consideration by *eLife*. Your revised article has been evaluated by Kenton Swartz as the Senior Editor and three peer reviewers, one of whom is a member of our Board of Reviewing Editors.

Your manuscript has been examined by three reviewers who have various degrees of expertise relevant to your study. As you will note from the detailed reviews below, the reviewers were highly enthusiastic about your new sensor and they feel that this manuscript would be a good fit for *eLife*.

However, they raise several points that need to be addressed in detail prior to any formal decision. First, reviewers 1 and 2 were surprised at the lack of technical detail in the manuscript concerning the development of the new sensor. Second, it was generally felt that the model was somewhat preliminary and certainly does not justify some of the conclusions concerning relevance to the fate of pyruvate in cells. While we will leave it up to you to decide how to approach this point, you may decide to leave the model out or as a minimum to minimize the interpretation of the modeling data. Third, was the issue of variability in your measures of pyruvate, particularly in mitochondria. Based on this variability one wonders what the level of accuracy is in the rate measures. This exposed a major point raised by all reviewers concerning the advantages the new probe has over its previous iteration. The authors should really take major strides to elaborate upon this issue in their revision.

Reviewer #1:

This methodological study presents a new fluorescent protein-based reporter for intracellular pyruvate. While this group has previously reported a FRET-based pyruvate sensor, this design based on a circularly permuted FP has a greatly improved dynamic range and smaller size that enables it to be used for new applications. Most importantly, it can be targeted to the mitochondrial matrix enabling comparative measurements of pyruvate concentration with the cytoplasm. The authors show that this reporter can be used for several applications, including: quantifying inhibitors of pyruvate transporters, modeling pyruvate flux in the mitochondrial compartment, detecting heterogeneity between mitochondria within an individual cell, and performing ex vivo tissue measurements. Overall I think this an exciting development, and the study makes clear that the reporter has very significant potential in these areas. My largest criticism of the study is that it is somewhat light on detail, particularly in the Materials and methods.

1) The reporter is presented with essentially no description of its design and engineering process. Typically, cpGFP-based reporters require substantial iteration or selection to arrive at a functional There should be at least some summary description of how this was achieved, as this can add to understanding how it functions.

2) I think the modeling approach in Figure 3 is quite interesting, but the model would need substantially more validation to make the claim that "The main finding of this study is that mitochondrial pyruvate lies in the low μM range, endowing the MPC with ultrasensitive control over anaplerosis". At a minimum, the PDH and PC rate parameters would need to be confirmed for this particular system.

3) UK5099 has substantial fluorescence in the GFP range at concentrations used. It is surprising to see that this did not cause a problem for the MPC inhibition experiments. Was the fluorescence corrected for? Other details of the computational image processing should also be included in the Materials and methods.

Reviewer #2:

Arce-Molina et al. present a manuscript in which they develop an improved pyruvate biosensor, which they subsequently use to characterize pyruvate transport and metabolism in the cytosol and mitochondria of mammalian cells.

Pyruvate is an important metabolite as end product of glycolysis, which can either be metabolized in the cytosol, or enter mitochondria via the mitochondrial pyruvate carrier, where it is used for example for oxidative phosphorylation.

The authors state that the main finding is ' that mitochondrial pyruvate lies in the low μM range, endowing the MPC with ultrasensitive control over anaplerotic processes.

As a first step, they found a suitable position for inserting cpGFP into an environmentally sensitive position of an allosterically regulated bacterial transcription factor. The sensor was characterized in vitro and in cell lines. Suitable control sensor variants and the less pH sensitive original Pyronic serve as suitable controls. A cytosolic version of the new sensor allows monitoring of pyruvate uptake into mammalian cells. Then the new Pyronic was targeted to the mitochondrial matrix and used to monitor uptake into mitochondria. Accumulation rates were slower compared to cytosolic uptake. Uptake was sensitive to inhibitors of MPC. Cytosolic levels were estimated to the low μM range in astrocytes. Using the data for parametrization of a mathematical model, regulatory aspects of MPC activity were predicted. It is noteworthy that the model applies to cell cultures grown under artificial conditions, and that while the calibration was elegant, a little more caution with the interpretation would be valuable. The authors then used an elegant approach, originally developed by them for estimating metabolic rates for glucose to estimate metabolism of pyruvate. Furthermore, first attempts were made to evaluate differences activities in individual mitochondria. As a next step, pyruvate was monitored in perineurial glia of *Drosophila* larvae.

Overall, this is an elegant study, in which the authors develop a new high sensitivity biosensors for pyruvate. It was not only used to measure uptake of pyruvate into mammalian cells, as well as *Drosophila* glia. Furthermore, the authors establish the sensors for monitoring MPC and metabolic activity. The sensors have tremendous potential for research and many medical applications.

- The authors found a suitable position in the allosterically regulated bacterial transcription factor, and provide the sequence in the supplement. It is however an achievement to find a suitable position for insertion of the cpGFP that ultimately provides a substrate-dependent change in fluorescence intensity. Thus the authors should provide more details on how they were able to develop such a sensor, which positions they tried, and how they arrived with the final version. Interesting is also to describe whether Pyronic still contains a DNA binding domain.

Reviewer #3:

In this study the authors have engineered a new pyruvate sensor and they conduct a series of experiments to validate the sensor and to demonstrate its utility for metabolic studies. The study is very thorough, and this sensor clearly improves on previous sensors.

1) There appears to be a much greater variance in cytosolic than mitochondrial pyruvate. Why do you think this might be? How does this level of variance compare with that of other metabolites? Do you have any way of proving that this biological rather than technical?

2) How confident are you that the probe is only found in the matrix. The pH of the matrix is higher than the cytosol. In view of the pH sensitivity of the probe how confident are you that the pH correction is accurate?

3) As the authors note the physiological concentration of pyruvate inside mitochondria is at the low end of the sensitivity curve for the sensor and one wonders if this is going to limit its utility for looking at things like physiological changes or mitochondrial variance. In Figure 3—figure supplement 4 it is noted that the top end of the concentration curve was truncated and a max was not reached which obscures the true sensitivity of the probe. Also how linear is the probe over what could be considered a physiological pyruvate concentration? It doesn't look that linear.

---

## [Author Response]

Reviewer #1:This methodological study presents a new fluorescent protein-based reporter for intracellular pyruvate. While this group has previously reported a FRET-based pyruvate sensor, this design based on a circularly permuted FP has a greatly improved dynamic range and smaller size that enables it to be used for new applications. Most importantly, it can be targeted to the mitochondrial matrix enabling comparative measurements of pyruvate concentration with the cytoplasm. The authors show that this reporter can be used for several applications, including: quantifying inhibitors of pyruvate transporters, modeling pyruvate flux in the mitochondrial compartment, detecting heterogeneity between mitochondria within an individual cell, and performing ex vivo tissue measurements. Overall I think this an exciting development, and the study makes clear that the reporter has very significant potential in these areas. My largest criticism of the study is that it is somewhat light on detail, particularly in the Materials and methods.

We thank the reviewer for her/his positive appraisal of this work. The Materials and methods section in the revised manuscript has been expanded as detailed below.

1) The reporter is presented with essentially no description of its design and engineering process. Typically, cpGFP-based reporters require substantial iteration or selection to arrive at a functional There should be at least some summary description of how this was achieved, as this can add to understanding how it functions.

The rationale and procedures for the selection of the sensor are now described in the Materials and methods section.

2) I think the modeling approach in Figure 3 is quite interesting, but the model would need substantially more validation to make the claim that "The main finding of this study is that mitochondrial pyruvate lies in the low μM range, endowing the MPC with ultrasensitive control over anaplerosis". At a minimum, the PDH and PC rate parameters would need to be confirmed for this particular system.

We concur. The conclusion at the end of the Introduction now states that the main finding is that mitocondrial pyruvate is low. The phrase regarding the MPC has been toned down and now refers to the potential capability of modulating anaplerosis. The need for PDH and PC parameters in astrocytes is now mentioned in the subsections “Astrocytic mitochondrial pyruvate is at optimal level for MPC modulation of anaplerosis” and “Ultrasensitive regulation of anaplerosis/gluconeogenesis by the MPC”.

3) UK5099 has substantial fluorescence in the GFP range at concentrations used. It is surprising to see that this did not cause a problem for the MPC inhibition experiments. Was the fluorescence corrected for? Other details of the computational image processing should also be included in the Materials and methods.

We could not find information about UK5099 fluorescence in the GFP range in the literature. However, the lack of detectable change in the fluorescence of extracellular ROIs upon addition of 10 μM UK-5099 (Author response image 1) implies that at least at this concentration autofluorescence is not a problem.

**Author response image 1. respfig1:** Lack of measurable fluorescence of 10 µM UK-5099. The response to 10 µM UK-5099 of an astrocyte expressing mito-PyronicSF was compared with that of a neighboring region of interest (ROI) devoid of cells (488 excitation, 525 ± 25 emission). While the astrocytic signal decreased steadily, there was no apparent change in the intensity of the extracellular ROI.

Reviewer #2:Arce-Molina et al. present a manuscript in which they develop an improved pyruvate biosensor, which they subsequently use to characterize pyruvate transport and metabolism in the cytosol and mitochondria of mammalian cells.Pyruvate is an important metabolite as end product of glycolysis, which can either be metabolized in the cytosol, or enter mitochondria via the mitochondrial pyruvate carrier, where it is used for example for oxidative phosphorylation.The authors state that the main finding is ' that mitochondrial pyruvate lies in the low μM range, endowing the MPC with ultrasensitive control over anaplerotic processes.As a first step, they found a suitable position for inserting cpGFP into an environmentally sensitive position of an allosterically regulated bacterial transcription factor. The sensor was characterized in vitro and in cell lines. Suitable control sensor variants and the less pH sensitive original Pyronic serve as suitable controls. A cytosolic version of the new sensor allows monitoring of pyruvate uptake into mammalian cells. Then the new Pyronic was targeted to the mitochondrial matrix and used to monitor uptake into mitochondria. Accumulation rates were slower compared to cytosolic uptake. Uptake was sensitive to inhibitors of MPC. Cytosolic levels were estimated to the low μM range in astrocytes. Using the data for parametrization of a mathematical model, regulatory aspects of MPC activity were predicted. It is noteworthy that the model applies to cell cultures grown under artificial conditions, and that while the calibration was elegant, a little more caution with the interpretation would be valuable. The authors then used an elegant approach, originally developed by them for estimating metabolic rates for glucose to estimate metabolism of pyruvate. Furthermore, first attempts were made to evaluate differences activities in individual mitochondria. As a next step, pyruvate was monitored in perineurial glia of *Drosophila* larvae.Overall, this is an elegant study, in which the authors develop a new high sensitivity biosensors for pyruvate. It was not only used to measure uptake of pyruvate into mammalian cells, as well as *Drosophila* glia. Furthermore, the authors establish the sensors for monitoring MPC and metabolic activity. The sensors have tremendous potential for research and many medical applications.

We thank the reviewer for her/his positive appraisal of this work. We are now more cautious about the limitations of the study.

- The authors found a suitable position in the allosterically regulated bacterial transcription factor, and provide the sequence in the supplement. It is however an achievement to find a suitable position for insertion of the cpGFP that ultimately provides a substrate-dependent change in fluorescence intensity. Thus the authors should provide more details on how they were able to develop such a sensor, which positions they tried, and how they arrived with the final version. Interesting is also to describe whether Pyronic still contains a DNA binding domain.

The rationale and procedures for the selection of the sensor are now described in the Materials and methods section. PyronicSF still contains the DNA binding domain of PdhR, this is now mentioned in the Results section (subsection “A highly responsive pyruvate sensor”).

Reviewer #3:In this study the authors have engineered a new pyruvate sensor and they conduct a series of experiments to validate the sensor and to demonstrate its utility for metabolic studies. The study is very thorough, and this sensor clearly improves on previous sensors.1) There appears to be a much greater variance in cytosolic than mitochondrial pyruvate. Why do you think this might be? How does this level of variance compare with that of other metabolites? Do you have any way of proving that this biological rather than technical?

Prompted by these questions, we have investigated the possible sources of heterogeneity (new Figure 3—figure supplement 2 and Results subsection “Quantification of mitochondrial pyruvate concentration”). Part of the variability is between culture plates, which may be due to animals and culture conditions. On top of that, there are differences between cells within the same plate. Indeed, astrocytic metabolic parameters may differ by a factor of 10 in a culture plate, for example glucose concentration, glycolytic rate, NADH/NAD+ ratio and lactate concentration (now cited). It will be great to measure multiple parameters simultaneously to correlate variance with cell cycle, degree of contact inhibition, etc.

2) How confident are you that the probe is only found in the matrix. The pH of the matrix is higher than the cytosol. In view of the pH sensitivity of the probe how confident are you that the pH correction is accurate?

Based on Figure 2—figure supplement 3B, a fast component in the fluorescence increase elicited by cell exposure to pyruvate indicates that some of the sensor is not in the mitochondrial matrix but in the cytosol. The same reasoning can be applied to the mitochondrial intermembrane space, which is freely connected to the cytosol. That is, a significant degree of sensor targeting to the intermembrane space should be detected as a fast component (e.g. such as cells 2 or 3 in Figure 2—figure supplement 3B). This is now discussed in the subsection “Estimation of MPC activity”. Fortunately, pH displaces the pyruvate dose response in parallel fashion, without affecting its kinetic parameters (Figure 1—figure supplement 2A). Therefore, the correction is pretty robust.

3) As the authors note the physiological concentration of pyruvate inside mitochondria is at the low end of the sensitivity curve for the sensor and one wonders if this is going to limit its utility for looking at things like physiological changes or mitochondrial variance. In Figure 3—figure supplement 4 it is noted that the top end of the concentration curve was truncated and a max was not reached which obscures the true sensitivity of the probe. Also how linear is the probe over what could be considered a physiological pyruvate concentration? It doesn't look that linear.

Even though pyruvate concentration inside mitochondria is at the low end of the sensitivity curve, its large fluorescence change endows PyronicSF with enough sensitivity to detect a 10 μM gap in the zero to 20 μM range, provided that sufficient number of replicates are available (Figure 3—figure supplement 2). Actually, it is more sensitive than the FRET sensor even in the low μM range (new Figure 3—figure supplement 5). For fast time courses, in which sampling is limiting, the sensor may not have sufficient sensitivity. This limitation is now discussed in the subsection “Quantification of mitochondrial pyruvate concentration”. The dose response at low pyruvate concentrations looks actually quite linear. This is now made more clear in the revised Figure 3—figure supplement 4), with a linear X axis. The range of the probe in the series of experiments of Figure 3—figure supplement 4) was confirmed by a measurement at 10 mM pyruvate, which resulted in a 188% change in fluorescence. This is now stated the legend to Figure 3—figure supplement 4.